# The Added Value of Systematic Sampling in In-Bore Magnetic Resonance Imaging-Guided Prostate Biopsy

**DOI:** 10.3390/jpm13091373

**Published:** 2023-09-14

**Authors:** Alon Lazarovich, Tomer Drori, Dorit E. Zilberman, Orith Portnoy, Zohar A. Dotan, Barak Rosenzweig

**Affiliations:** 1Department of Urology, Chaim Sheba Medical Center, Ramat Gan 5266202, Israel; alon.lazarovich@gmail.com (A.L.); tomer.drori@sheba.health.gov.il (T.D.); dorit.zilberman@sheba.health.gov.il (D.E.Z.); zohar.dotan@sheba.health.gov.il (Z.A.D.); 2The Sackler Faculty of Medicine, Tel Aviv University, Tel Aviv 6997801, Israel; orith.portnoy@sheba.health.gov.il; 3Department of Diagnostic Imaging, Chaim Sheba Medical Center, Ramat Gan 5266202, Israel

**Keywords:** systematic biopsy, multi-parametric magnetic resonance imaging, magnetic resonance-guided prostate biopsy, magnetic resonance-guided systematic biopsy, in-bore magnetic resonance-guided prostate biopsy

## Abstract

**Simple Summary:**

We quantified the additive diagnostic value of systematic biopsy (SB) using in-bore magnetic resonance-guided prostate biopsy (IBMRGpB) by retrospectively reviewing 189 records of patients who were suspected to have prostate cancer or were previously diagnosed with prostate cancer. Sixty-seven (35.5%) patients had positive findings. SB from the lobe contralateral to the lesion detected clinically significant disease in 15 (12%) patients. The prostate size was larger and the number of lesions in the peripheral zone was greater in men with positive SB findings compared to those with negative SB findings. Main lesions located in the peripheral zone of the prostate were predictive of clinically significant disease, a finding supported by a subgroup analysis of biopsy-naïve patients. Adding SB to IBMRGpB increased the capability of diagnosing both clinically significant and non-clinically significant prostate cancer. These findings may enhance patient-tailored management.

**Abstract:**

We sought to quantify the additive value of systematic biopsy (SB) using in-bore magnetic resonance (MR)-guided prostate biopsy (IBMRGpB) by retrospectively reviewing the records of 189 patients who underwent IBMRGpB for suspected prostate cancer or as part of the surveillance protocol for previously diagnosed prostate cancer. The endpoints included clinically significant and non-clinically significant cancer diagnosis. SB detected clinically significant disease in 67 (35.5%) patients. Five (2.65%) patients whose targeted biopsies indicated benign or non-clinically significant disease had clinically significant disease based on SB. SB from the lobe contralateral to the lesion detected clinically significant disease in 15 (12%) patients. The size of the prostate was larger and the percentage of lesions located in the peripheral zone of the prostate was higher in patients with SB-detected clinically significant disease. The location of the main lesion in the peripheral zone of the prostate was a predictor for clinically significant disease in the multivariate analysis (OR = 8.26, *p* = 0.04), a finding supported by a subgroup analysis of biopsy-naïve patients (OR = 10.52, *p* = 0.034). The addition of SB during IBMRGpB increased the diagnosis of clinically significant as well as non-clinically significant prostate cancer. The location of the main lesion in the peripheral zone emerged as a positive predictive factor for clinically significant disease based on SB. These findings may enhance patient-tailored management.

## 1. Introduction

Multi-parametric magnetic resonance imaging (mpMRI) has now become the leading tool for diagnosing clinically significant prostate cancer, while a magnetic resonance-guided prostate biopsy (MRGpB) is considered superior to a transrectal ultrasound (TRUS)-guided biopsy for the purpose of histologic diagnosis [1,2,3]. The combination of mpMRI to locate and define suspected lesions, together with them being targeted by MRGpB, has succeeded in increasing the rate of detection of clinically significant disease and thereby lowering the detection of non-significant prostate cancer [4,5]. Such data were acquired in randomized controlled trials which compared systematic biopsies (SBs) guided by TRUS to MRGpBs combining lesion-directed biopsy and SB (i.e., ultrasound/magnetic resonance imaging [US/MRI] fusion). Although recent data challenge the conclusion that MRGpB combining SB is superior to SB guided by TRUS [6], US/MRI fusion biopsy including systematic sampling is still routinely performed by urologists worldwide. Other investigators found that the addition of SB (MRGSB) to MRGpB increased the detection rate of clinically significant prostate cancer compared to MRGpB alone [7,8,9,10]. 

A multicenter randomized controlled trial (the FUTURE trial) compared overall and clinically significant prostate cancer detection rate between three transrectal MRGpB methods: cognitive, fusion and in-bore. There were no significant differences in detection rates of overall and clinically significant prostate cancer. However, this study was underpowered, and the overall prostate cancer detection rate was higher with the in-bore method (cognitive 44%, fusion 49% and in-bore 55%), which might be considered a more accurate sampling technique [11]. As such, it can potentially reveal the actual benefit of adding systematic to targeted sampling. In this work, we sought to determine the additive value of systematic sampling to biopsies targeted by in-bore MRGpB.

## 2. Materials and Methods

### 2.1. Ethics and Patient Cohort

Following approval by the Institutional Review Board for a waiver of informed consent, we retrospectively collected data of 189 patients who underwent IBMRGpB in a tertiary center between 2017 and 2022. The patients were referred for biopsy for suspected prostate cancer or as part of the surveillance protocol for previously diagnosed prostate cancer. Patients who were suspected to have prostate cancer had elevated PSA (prostate-specific antigen) serum levels and/or abnormal digital rectal examination (DRE), and ≥1 suspicious areas on mpMRI scan with a score of ≥3 according to the Prostate Imaging Reporting and Data System (PIRADS v.2). All study patients had serum PSA levels ≤20 ng/mL and more than eight cores were taken at systematic sampling.

All patients enrolled in this study had SBs taken with the intention of performing a formal 12-core template consisting of two samples (one on each side) from the prostate base, with two from the mid-gland and two from the apex. All biopsies were performed transrectally following the administration of prophylactic antibiotics (Ciprofloxacin 500 mg twice daily for 5 days before the procedure, and Gentamycin 240 mg 30 min before the procedure).

The in-bore MRGpBs were carried out by means of 3T MRI scanners and external coil application. Imaging during the biopsies included T2-weighted images and diffusion series. The IBMRGpB patients were placed in a prone position and administered general anesthesia. A digital rectal exam was performed to determine if there were any anatomic or pathological conditions that could hinder transrectal biopsy and to approximate the position of the gland. Axial and sagittal T2-weighted images were obtained to visualize the prostate and identify the target lesion. A nonmagnetic portable biopsy device (DynaTRIM; Invivo, Gainesville, FL, USA) and a dedicated software package for device tracking and target localization (DynaCAD; Invivo) were also used. Suspected clinically significant target lesions that were detected by MRI were sampled first, followed by SB using the last MRI acquired to mark needle coordinates for all twelve cores.

IBMRGpBs were carried by a team of dedicated uro-radiologists with over 10 years of experience in prostate MRI reading, an anesthesiologist, a single fellowship-trained urologic oncologist, and an experienced nurse and technician. The biopsy specimens were processed by means of routine pathologic fixation with formalin solution and evaluated by a single dedicated uropathologist with over 20 years of experience. The retrieved cancer cells were used as the reference standard to determine the positivity of the biopsy. Both Gleason grade (6–10) and ISUP (International Society of Urological Pathology) scale (1–5) used for prostate cancer histologic grading were depicted in the pathology report. We stratified the results according to levels of clinically significant disease, defined as a Gleason score of ≥7 (ISUP ≥ 2), and non-clinically significant disease, defined as a Gleason score of 6 (ISUP 1).

### 2.2. Statistical Analysis

The data were analyzed through descriptive statistics. Comparison between groups was performed using Pearson’s Chi-squared test for qualitative variables and Fisher’s exact test for dichotomous variables. Comparison of quantitative variables was performed using parametric *t*-tests and the non-parametric Mann–Whitney test. A multivariate analysis using logistic regression was performed to identify indicators that predicted the presence of clinically significant disease in the SBs. 

Given that focal therapies are often aimed at different anatomic areas of a single lobe, we separately analyzed the results of the SBs from the contralateral lobe to the region of interest (ROI), including those of patients who had suspected lesions in only one lobe. The data are presented as an average (standard deviation), a median (interquartile range [IQR]) or a numerical value (relative share, %). The level of statistical significance was defined as *p* < 0.05. All statistical analyses were performed using SPSS version 21, IBM corporation.

## 3. Results

Our retrospective search of the medical center’s database yielded 189 patients who underwent IBMRGpB by a single uro-oncologist in our institution during the study period. The cohort’s median age was 68 years (IQR 62–71.5), the mean prostate size as calculated using MRI was 56.8 ± 35.3 cc, and the mean PSA and PSA density levels were 8.7 ± 8.2 ng/mL and 0.2 ± 0.2 ng/mL/cc, respectively. Forty-two patients (22.2%) had suspicious DRE findings. We doubled the PSA values of the 15 (7.9%) patients who used 5-alpha-reductase inhibitors (Table 1). A total of 113 (59.8%) patients had undergone a previous prostate biopsy, of whom 60 (31.7%) and 49 (25.9%) had a benign histology and an International Society of Urological Pathology grade of 1 (ISUP1), respectively. Four patients (2.1%) with minimal findings of ISUP2 disease opted for active surveillance (AS). The median number of cores taken from the ROI was 5 (IQR 4–7), and the median total of cores per procedure was 17 (IQR 16–19). We found a total of 287 clinically significant lesions on mpMRI. Any prostate cancer (clinically and non-clinically significant) was found in 162 (56.4%) lesions: 33 (31%), 100 (68%) and 29 (88%) were categorized as PIRADS 3, 4 and 5, respectively. Clinically significant prostate cancer was found in 15 (14%), 69 (47%) and 25 (76%) lesions categorized as PIRADS 3, 4 and 5, respectively (Figure 1). The group of patients with clinically significant disease as identified in the SB samples (*n* = 67) was compared to the group of patients with non-clinically significant disease (*n* = 122). The patients’ characteristics are depicted in Table 1.

An examination of the SBs found clinically significant disease in 67 (36%) patients overall. SBs from the lobe contralateral to the lesion detected clinically significant disease in 15 (12%) patients (124 patients [65.6%] had lesions in one prostate lobe and were eligible for the analysis, Figure 2). Four (2.1%) patients had clinically significant disease based on their SB while their samples from the ROI were benign. One patient (0.5%) had clinically significant disease on his SB while his samples from the ROI indicated non-clinically significant disease. Finally, 36 (19%) patients had clinically significant disease on both their SB and their ROI.

An examination of the SBs found non-clinically significant disease in 119 (63%) patients. SBs from the contralateral lobe revealed non-clinically significant disease in 27 (21.8%) patients. Eleven (5.8%) patients had non-clinically significant disease based on their SB and the samples from their ROI were benign. The multivariate analysis revealed that the location of the main lesion in the peripheral zone (PZ) was a predictor for finding clinically significant disease in the SB samples (odds ratio [OR] = 8.26, *p* = 0.04).

A subgroup analysis of biopsy-naïve patients (*n* = 73) found that patients with clinically significant disease on their SB had a higher percentage of suspicious DRE, a higher percentage of 5-alpha-reductase therapy, a lower prostate size (as calculated using MRI), and a higher percentage of the main lesions located in the PZ of the prostate (Table 2). The multivariate analysis determined that the location of the main lesion in the PZ was a predictor for finding clinically significant disease in the SB samples (OR = 10.52, *p* = 0.034).

## 4. Discussion

This was a retrospective analysis of data from 189 patients who underwent IBMRGpB at our institution during the study period. Its purpose was to assess the added value of SB for targeted biopsies during IBMRGpB and identify potential predictive factors to support the additive benefit of this sampling methodology. The detection rate of clinically significant prostate cancer based on lesions categorized as 3, 4 and 5 in our cohort according to the PIRADS version 2 was 14%, 47% and 76%, respectively. These results are in line with those reported in the literature [1,12], thereby validating the accuracy of our urologists’ MRI interpretation and analysis of lesion sampling.

In total, 119 (63%) of the patients in our cohort had non-clinically significant disease and 67 (35.5%) had clinically significant disease based on the SB sampling. The rate of clinically significant disease found using SB during IBMRGpB in our cohort was higher than the rate described for SB during fusion MRGSB by Kim et al. (27%) [13] and Hanna et al. (24.7%) [9]. The differences between the cohorts might be due to the higher rate of patients who had negative previous biopsy in their [9] cohort (46.4% vs. 31.7% in our cohort). The relatively higher mean serum PSA level and the higher percentage of suspicious DRE in our cohort compared to that of Kim et al. and Hanna et al. also support our findings. However, the rate of our biopsy-naïve patients was higher than that of Hanna et al. (38.6% vs. 15.2%). In contrast to the above findings, an earlier work by our group found no difference in SBs when comparing those performed using TRUS to those performed under IBMRGpB [14].

Many practitioners believe that the desired achievement in the diagnosis of prostate cancer is to identify patients with clinically significant disease and to avoid the diagnosis of non-clinically significant cancer. On the one hand, we found clinically significant disease based on the SBs of five patients (2.65%) who had non-clinically significant or benign disease in the ROI. On the other hand, we found non-clinically significant disease based on the SBs of 11 patients (5.8%) who had benign disease in the ROI. In an attempt to find predictive factors for diagnosing clinically significant disease using SB, we found that the location of the main lesion in the PZ of the prostate to be a positive prognostic factor (OR = 8.26, *p* = 0.04, Table 1). A subgroup analysis of 73 biopsy-naïve patients further supported this observation (OR = 10.52, *p* = 0.034, Table 2). Our finding is suggestive of a risk-stratifying factor that may assist in avoiding unnecessary SB sampling in selected cases while being indicative of such an approach in other cases.

With an increasing awareness of the possible harm of radical prostate cancer treatments, AS has been suggested for men with ISUP1 and some low-volume ISUP2 prostate cancer. However, not all patients consent to this approach, and, moreover, outcomes for AS in ISUP2 patients are reportedly not as good as those for ISUP1 [15]. Treatment alternatives aiming to improve functional outcomes include focal therapy, and although the ideal candidate for such treatment is a matter of debate [16,17], the choice of this surgical approach reflects the need to ensure that there is no cancer outside of lesions seen on mpMRI. Many focal therapy treatments include multiple forms of hemi-ablations [18,19]. We therefore analyzed the SB samples in the lobe contralateral to the ROI in patients who had suspicious lesions in a single prostate lobe and found both non-clinically significant and clinically significant disease in 27 (21.8%) and 15 (12%) patients, respectively. These results might affect the decision regarding the extent of focal therapy coverage (lesion-only, hemi-ablations, hockey stick, etc.) and lead to the provision of radical therapy in some patients who might otherwise be scheduled to undergo focal therapy.

The main limitations to this study are its retrospective design and the fact that the data were derived from a single institution. We also enrolled a diverse population of patients, including some patients who were biopsy-naïve, some who had undergone previous prostate biopsy, and some who had been diagnosed with prostate cancer and were allocated to AS. Such diversity reflects the nature of our patients who were referred for biopsy with or without SB sampling. The fact that our subgroup analysis of biopsy-naïve patients yielded similar results supports the generalizability of our findings. Finally, the procedures we describe were performed by a highly trained uro-oncologist and a team of dedicated genito-urinary radiologists in a tertiary center, and our results might not be applicable to less specialized centers.

In summary, the results of our study demonstrate that the addition of SB to IBMRGpB, which is highly accurate, increases the diagnostic accuracy of clinically significant prostate cancer, as 35.5% of the patients had clinically significant disease based on SB sampling. However, only 2.65% of the patients had clinically significant disease based on SB sampling while their targeted biopsy found benign or non-clinically significant disease. This is a fact that may weaken the claim that SB sampling is still mandatory during IBMRGpB. SB found clinically significant disease in the contralateral lobe to the ROI in 12% of the patients, a fact that further supports the importance of SB in an era when interest in focal therapy rises, and identifying clinically significant disease outside of the ROI may affect treatment plans. Moreover, we demonstrated that the location of the main lesion in the PZ correlated positively with clinically significant disease based on SB, and we suggest that this result may serve as a means for stratifying risk, thereby encouraging the addition of SB during IBMRGpB.

## 5. Conclusions

Adding SB to IBMRGpB increases the diagnosis of both clinically and non-clinically significant prostate cancer and might affect focal therapy treatment plans. The location of the main lesion in the PZ of the prostate is a positive predictive factor for clinically significant disease based on SB.

## Figures and Tables

**Figure 1 jpm-13-01373-f001:**
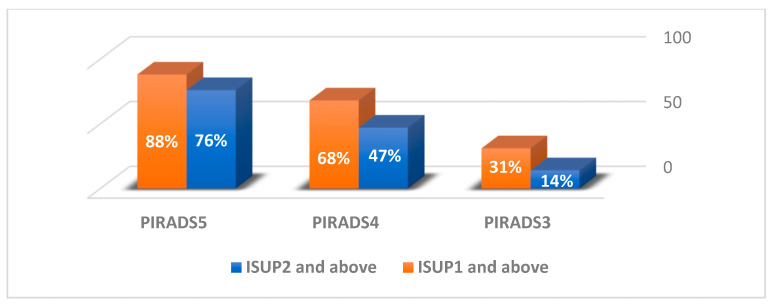
In-bore mpMRI biopsy pathological results of 287 mpMRI lesions by PIRADS version 2 score. PIRADS = Prostate Imaging Reporting and Data System. ISUP = International Society of Urological Pathology.

**Figure 2 jpm-13-01373-f002:**
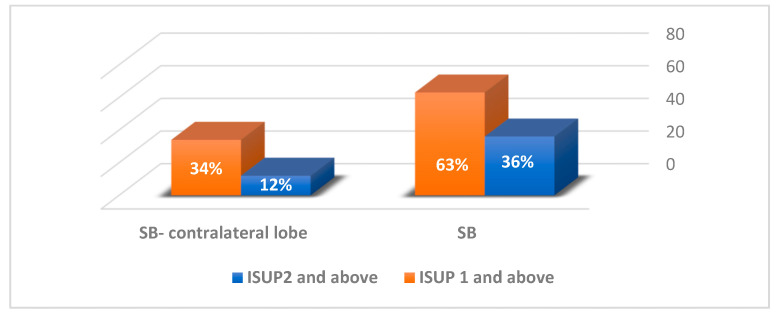
SB pathological results of the whole cohort (189 patients) and of the patients who had lesions in one prostate lobe and were eligible for analysis of SB from the contralateral lobe (124 patients). SB = systematic biopsy. ISUP = International Society of Urological Pathology.

**Table 1 jpm-13-01373-t001:** Patient characteristics.

Variable	All Patients No. %		Clinically Significant on SB No. %		Non-Clinically Significant on SB No. %		*p* Value ^3^
Age, years (IQR)	68	(62–71.5)	68	(64–73)	67	(61–70)	NS
Prostate size, cc ^1,2^	56.8	35.3	49.6	29.7	60.7	37.6	**0.05**
Pre-biopsy PSA, ng/mL ^1^	8.7	8.2	8.8	5.9	8.7	9.3	NS
PSAD, ng/mL/cc ^1^	0.2	0.27	0.23	0.24	0.18	0.29	NS
DRE							NS
Non-suspicious (T1c)	139	73.5	44	65.7	95	77.9	
Suspicious (T2a)	42	22.2	21	31.3	21	17.2	
NA	8	4.2	2	3	6	4.9	
Previous biopsy							NS
No	73	38.6	31	46.3	42	34.4	
Yes	113	59.8	34	50.7	79	64.8	
NA	3	1.6	2	3	1	0.8	
5-alpha-reductase inhibitor							NS
No	169	89.4	60	89.6	109	89.3	
Yes	15	7.9	5	7.5	10	8.2	
NA	5	2.6	2	3	3	2.5	
LND involvement on MRI							NS
No	132	69.8	48	71.6	84	68.9	
Yes	5	2.6	1	1.5	4	3.3	
NA	52	27.5	18	26.9	34	27.9	
Main lesion location							**0.008 ^4^**
PZ	136	72	58	86.6	78	63.9	
TZ	39	20.6	6	9	33	27	
PZ + TZ	1	0.5	0	0	1	0.8	
NA	13	6.9	3	4.5	10	8.2	

^1^ Values are presented as average and standard deviation. ^2^ Measured using MRI. ^3^
*p*-value, univariate analysis. ^4^ Statistically significant based on multivariate analysis. SB, systematic biopsy; IQR, interquartile range; PSA, prostate-specific antigen; PSAD, PSA density (calculated according to prostate size as measured using MRI); DRE, digital rectal exam; LND, lymph node; PZ, peripheral zone; TZ, transitional zone; NA, not available. Bold indicates significant.

**Table 2 jpm-13-01373-t002:** Characteristics of biopsy-naïve patients.

Variable	All Patients No. %		Clinically Significant on SB No. %		Non-Clinically Significant on SB No. %		*p* Value ^3^
Age, years (IQR)	65.6	(61.5–72)	67.42	(64–73)	64.3	(57.7–71.2)	NS
Prostate size, cc ^1,2^	54	39.7	43.6	25.76	61.9	46.5	**0.045**
Pre-biopsy PSA, ng/mL ^1^	8.35	9.2	9.4	7.5	7.58	10.3	NS
PSAD, ng/mL/cc ^1^	0.246	0.38	0.28	0.3	0.21	0.46	NS
DRE							**0.05**
Non-suspicious (T1c)	55	75.3	19	61.3	36	85.7	
Suspicious (T2a)	16	21.9	11	35.5	5	11.9	
NA	2	2.7	1	3.2	1	2.4	
5-alpha-reductase inhibitor							**0.02**
No	69	95.8	27	87	42	100	
Yes	3	4.1	3	9.7	0	0	
NA	1	0.1	1	3.3	0	0	
LND involvement on MRI							NS
No	50	68.5	24	77.4	26	61.9	
Yes	3	4.1	1	3.2	2	4.8	
NA	20	27.4	6	19.4	14	33.3	
Main lesion location							**0.002 ^4^**
PZ	54	74	29	93.5	25	59.52	
TZ	14	19.2	1	3.2	13	30.95	
NA	5	6.8	1	3.3	4	9.52	

^1^ Values are presented as average and standard deviation. ^2^ Measured using MRI. ^3^
*p*-value, univariate analysis. ^4^ Statistically significant based on multivariate analysis. SB, systematic biopsy; IQR, interquartile range; PSA, prostate-specific antigen; PSAD, PSA density (calculated according to prostate size as measured using MRI); DRE, digital rectal exam; LND, lymph node; PZ, peripheral zone; TZ, transitional zone; NA, not available. Bold indicates significant.

## Data Availability

The data presented in this study are available from the corresponding author upon request. The data are not publicly available due to privacy policy.

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
