# Peer review of "The Added Value of Systematic Sampling in In-Bore Magnetic Resonance Imaging-Guided Prostate Biopsy"

_jpm, 2023, doi:10.3390/jpm13091373_

Round 1

Reviewer 1 Report

So, the 189 patients that has been reviewed were not confirmed to have prostate cancer, they were suspected to have prostate cancer? This needs to be clarified!

Redundancy in the two sentences presented in the simple summary” Main

lesions located in the peripheral zone of the prostate were predictive for clinically significant disease” and “The location of the main lesion in the peripheral zone was a positive predictive factor

for clinically significant disease”

Multi cited reference can be typed [1-3] instead of [1][2][3] and in case of two cited reference it can be typed [1,2]  

PSA is an acronym that should be spelled at first mention. Gleason score should have been mentioned in section 2.1. Further, a score 3 and 7 were used, this needs clarification.

All of the patients enrolled in the study instead of all of the study patients!!

Rewrite the sentence so it reads clear “and diffusion

series were used as necessary according to the radiologist’s discretion”

Any overlap between biopsy targeted lesion detected by MRI and the SB?

This should be moved to the result section or at least remove the n=67 and n=122.”The group of patients with clinically significant

disease in the SB samples (n=67) was compared to the group of patients with nonclinically

significant disease (n=122).

This reads confusing “Any prostate cancer was found in 162 (56.4%) lesions: 33 (30.8%), 100 (68%) and

29 (87.9%) were categorized as PIRADS 3, 4 and 5, respectively. Clinically significant

prostate cancer was found in 15 (14%), 69 (46.9%) and 25 (75.7%) lesions categorized as

PIRADS 3, 4 and 5, respectively.”

One patient (0.5%) had clinically significant disease on SB while his samples

from the ROI indicated non-clinically significant disease. Does this mean the positive lesion was missed since the histology reflects non-significant disease? Elaborate more on the spatial scale used i.e., the distance between biopsies.

Biopsy could result in inflammation, how did this affect your finding particularly with people how been subjected to previous biopsy?

Tables 1 & 2 were not referred to throughout the manuscript i.e., not cited.

The manuscript should be considered for professional English language editing. 

Author Response

Reviewer #1:

1. So, the 189 patients that has been reviewed were not confirmed to have prostate cancer, they were suspected to have prostate cancer? This needs to be clarified!

This point was edited and clarified in the simple summary (page 1), abstract (page 1) and Methods -Ethics and patient cohort (page 2).

2. Redundancy in the two sentences presented in the simple summary- "Main lesions located in the peripheral zone of the prostate were predictive for clinically significant disease” and “The location of the main lesion in the peripheral zone was a positive predictive factor for clinically significant disease".

The sentences " The location of the main lesion in the peripheral zone was a positive predictive factor for clinically significant disease" was deleted from the simple summary.

3. Multi cited reference can be typed [1-3] instead of [1][2][3] and in case of two cited reference it can be typed [1,2].

Corrected in the text.

4. PSA is an acronym that should be spelled at first mention. Gleason score should have been mentioned in section 2.1. Further, a score ≥3 and ≥ 7 were used, this needs clarification.

PSA was spelled as "prostate specific antigen" in Methods- Ethics and patient cohort (page 2).

We elaborated regarding Gleason grade and ISUP in section 2.1 (page 2)- Both Gleason grade (6-10) and ISUP (International society of uropathology) scale (1-5) used for prostate cancer histology grading were the depicted in the pathology report. We stratified the results according to levels of clinically significant disease defined as a Gleason score of ≥7 (ISUP ≥2) and non-clinically significant disease defined as a Gleason score of 6 (ISUP 1).

5. All of the patients enrolled in the study instead of "All of the study patients".

The sentence was edited as requested (section 2.1, page 2).

6. Rewrite the sentence so it reads clear “and diffusion series were used as necessary according to the radiologist’s discretion”.

We edited the sentence -   Imaging during the biopsies included T2-weighted images and diffusion series (section 2.1, page 2).

7. Any overlap between biopsy targeted lesion detected by MRI and the SB?

The SB were done following the targeted biopsies detected by the MRI on the same session. The SB were "targeted" outside of the lesions detected by the MRI and the coordinates for the SB were measured.

8. This should be moved to the result section or at least remove the n=67 and n=122. ” The group of patients with clinically significant disease in the SB samples (n=67) was compared to the group of patients with non-clinically significant disease (n=122).”

The sentence was removed to the end of paragraph 1 of the result section, page 3.

9. This reads confusing “Any prostate cancer was found in 162 (56.4%) lesions: 33 (30.8%), 100 (68%) and 29 (87.9%) were categorized as PIRADS 3, 4 and 5, respectively. Clinically significant prostate cancer was found in 15 (14%), 69 (46.9%) and 25 (75.7%) lesions categorized as PIRADS 3, 4 and 5, respectively.”

"Any prostate cancer" means clinically and non – clinically significant prostate cancer. A clarification was made in the text. Paragraph 1 of the result section, page 3.

 10. One patient (0.5%) had clinically significant disease on SB while his samples from the ROI indicated non-clinically significant disease. Does this mean the positive lesion was missed since the histology reflects non-significant disease? Elaborate more on the spatial scale used i.e., the distance between biopsies.

No. That means that the SB found clinically significant disease outside of the targeted lesion. " Suspected clinically significant target lesions that were detected by MRI were sampled first, followed by SB using the last MRI acquired to mark needle coordinates for all twelve cores" (section 2.1, page 2).

11. Biopsy could result in inflammation; how did this affect your finding particularly with people now been subjected to previous biopsy?

Unfortunately, we don't have the exact answer for that question, as we did not compare the incidence of inflammation in patients with previous prostate biopsies and biopsy naïve patients. However, most of the patients with previous biopsy had ISUP 1 and were under active surveillance (AS) protocol or had BPH. Which means that the previous biopsy was done at least a few months before the current biopsy and did not have a major effect on the current biopsy results. In our institution a repeated biopsy during AS is planned every 3 years or when progression occurs, e.g. a new finding in the mpMRI (performed every 1.5 years).

12. Tables 1 & 2 were not referred to throughout the manuscript i.e., not cited.

Table 1 and 2 are now cited in the results section, pages 3 and 4.

Reviewer 2 Report

Overall, the paper is well put together, bringing at the fore-front an up-to-date topic. Additionally, the number of patients included is impressive, given the employed biopsy technique.

The introduction could potentially benefit from the following suggestions:

-  Please consider defining here what clinically significant and insignificant prostate cancer means.

-        I suggest that the authors discuss briefly the PCa detection rate and accuracy of each MRI-guided prostate biopsy approach: cognitive, fusion and in-bore.

The Results and the Conclusion need to address more clearly if systematic sampling is necessary in patients undergoing in-bore MRI-guided transrectal prostate biopsy. The authors have described the detection rate of PCa through systematic biopsies in the contralateral lobe and have quantified the number of cases that PCa was detected solely on the systematic samples, but a clear result of this research is needed.

Overall, the use of English language is adequate. Minor grammar mistakes and improper topic of the sentence can be spotted in the Introduction and Discussion sections.

Author Response

Reviewer #2:

1. The introduction could potentially benefit from the following suggestions:

Please consider defining here what clinically significant and insignificant prostate cancer means.

We defined clinically and non-clinically significant prostate cancer according to the comments of reviewer 1 in the end of section 2.1, page 3.

I suggest that the authors discuss briefly the PCa detection rate and accuracy of each MRI-guided prostate biopsy approach: cognitive, fusion and in-bore.

We added a short discussion regarding PCa detection rate and accuracy of each MRI-guided prostate biopsy approach in the end of the introduction paragraph, page 2.

2. The Results and the Conclusion need to address more clearly if systematic sampling is necessary in patients undergoing in-bore MRI-guided transrectal prostate biopsy. The authors have described the detection rate of PCa through systematic biopsies in the contralateral lobe and have quantified the number of cases that PCa was detected solely on the systematic samples, but a clear result of this research is needed.

We edited the final paragraph of the discussion section, page 5:
For summary, the results of our study demonstrated that the addition of SB to the highly accurate IBMRGpB increases the diagnostic accuracy of clinically significant prostate cancer, as 35.5% of the patients had clinically significant disease in SB sampling. However, only 2.65% of the patients had clinically significant disease in SB sampling while the targeted biopsy found benign or non-clinically significant disease. A fact which may weaken the claim that SB sampling is still mandatory during IBMRGpB.  SB found clinically significant disease in the contralateral lobe to the ROI in 12% of the patients, a fact which further supports the importance of SB in an era when focal therapy rises, and identifying clinically significant disease outside of the ROI may affect treatment plans. Moreover, we demonstrated that the location of the main lesion in the PZ correlated positively with clinically significant disease on SB and suggest that this result may serve as a means for stratifying risk, thereby encouraging the addition of SB during IBMRGpB.

Round 2

Reviewer 1 Report

Authors addressed the majority of the reviewer comments.  

I accept this work for publication after the 2nd review.